# Neurotransmitters in Prevention and Treatment of Alzheimer’s Disease

**DOI:** 10.3390/ijms24043841

**Published:** 2023-02-14

**Authors:** Zhenqi Yang, Yong Zou, Lifeng Wang

**Affiliations:** Beijing Institute of Radiation Medicine, 27 Taiping Road, Beijing 100850, China

**Keywords:** AD, neurotransmitter, receptors, target drugs

## Abstract

Alzheimer’s disease (AD) is the most frequent cause of cognitive impairment in middle-aged and older populations. There is a lack of drugs that demonstrate significant efficacy in AD, so the study of the pathogenesis of AD is of great importance. More efficacious interventions are needed, as reflected by our population’s fast aging. Synaptic plasticity is the capacity of neurons to adjust their connections, and it is strongly tied to learning and memory, cognitive function, and brain injury recovery. Changes in synaptic strength, such as long-term potentiation (LTP) or inhibition (LTD), are thought to represent the biological foundation of the early stages of learning and memory. The results of numerous studies confirm that neurotransmitters and their receptors play an important role in the regulation of synaptic plasticity. However, so far, there is no definite correlation between the function of neurotransmitters in aberrant neural oscillation and AD-related cognitive impairment. We summarized the AD process to understand the impact of neurotransmitters in the progression and pathogenesis of AD, including the current status of neurotransmitter target drugs, and the latest evidence of neurotransmitters’ function and changes in the AD process.

## 1. Alzheimer’s Disease’s Histopathological Variation

Epidemiological surveys have revealed that the prevalence of cognitive dysfunction in adults aged 65 and older is close to 40%, and that the prevalence grows exponentially with age. AD is a progressive neurodegenerative illness, characterized by extensive brain atrophy, neuronal cell death, neurogenic fiber tangles, and protein amyloidosis. Eighty-seven percent of AD patients have cognitive impairment before the onset of dementia symptoms. Researchers found that the main cause of the cognitive deficit that characterizes AD and aging was the loss of neurons and the hypothesis of an imbalance in the cellular and molecular mechanisms of synaptic plasticity underlying this deficit is currently widely accepted. Patients acquire cognitive dysfunction and secondary lesions in the neurological, urinary, and motor systems as AD develops, resulting in gradual behavioral loss. It is a progressive neurodegenerative disease, characterized by the loss of memory, multiple cognitive impairments and changes in the person’s personality and behavior [1]. Several hypotheses have been proposed to explain the pathogenesis of AD, including mitochondrial abnormalities and inflammatory responses, and the hypotheses of the accumulation of amyloid-beta (Aβ) and phosphorylated tau are widely accepted. The amyloid precursor protein (APP) gene encodes Aβ, an evolutionarily conserved protein generated during normal brain metabolism. APP is an integral type I transmembrane protein with a large extra-cellular glycosylated N-terminal domain and a shorter cytoplasmic C-terminal domain, which is critical for neuronal development and function [2]. Physiologically, CNS can maintain the balance of Aβ synthesis and clearance, but when it collapses, for example, the modification enzyme cleaves at the incorrect place, and causes alterations in the sequence and structure of Aβ and the deposition of large amounts of amyloid in the interstitial space of neurons will form the classic pathological feature of AD, Aβ amyloid plaques. There are three forms Aβ protein, which are as follows: soluble Aβ protein, oligomeric Aβ protein, and Aβ amyloid plaques. These three variants of Aβ proteins may cause neuronal death via diverse mechanisms, including mitochondrial apoptosis, neuroinflammatory response, and oxidative stress response [3]. Soluble Aβ has been shown in studies to cause more serious pathological damage to neurons; soluble Aβ may bind to neurosynaptic receptors, such as NMDAR (N-methyl-D-aspartic acid receptor), to affect their normal signaling, cause continuous calcium ion inward flow and neuronal hyperactivation, and form a vicious cycle that eventually leads to synaptic dysfunction, inflammation and cell death [4]. NFTs (neurofibrillary tangles) are primarily intra-neuronal filamentous inclusions and consist of aggregated abnormal hyperphosphorylated tau proteins [5]. Tau tangles, the hallmark of AD pathology, can bind to synaptic vesicle surface proteins, causing increased vesicle aggregation at the synapse’s front, limited movement, and decreased release [6]. The tau protein is a kind of microtubule-associated phosphoprotein predominantly expressed in neuronal cells, mainly in axons, which is essential for the proper assembly, stabilization, and functioning of a microtubule network. Tau also regulates axonal transport, drives neurite outgrowth, and shapes the neuronal morphology of aberrantly misfolded and hyperphosphorylated tau proteins [7]. A synapse is the basic unit of information transfer and functional connecting point between neurons. Hyperphosphorylated tau proteins in neurons can cause neural dysfunction by interfering with microtubule stabilization and axonal transport [8]. As an axonal cytoskeletal protein, synaptic tau engages in neuronal signaling and synaptic plasticity under physiological conditions, which is produced in neurons and attaches to axonal microtubules. However, under pathological situations, mutations in tau or affinity alterations result in tau separation from axonal microtubules and their subsequent mis-localization to synapses, interfering with intrasynaptic material transfer, and the formation of protein hydrolase-insensitive neurofibrillary hyperphosphorylation in neuronal axons [6]. Hyperphosphorylated tau is an essential pathological characteristic of AD, and soluble Aβ protein can increase fibrillary hyperphosphorylation development by promoting tau phosphorylation [9]. Tau has both soluble and polymeric phases, and soluble tau can cause neurotoxicity and impair neuronal function [10]. The existing research has generally concluded that Aβ amyloid and tau hyperphosphorylation are relatively independent pathological changes, with soluble Aβ causing neuronal hyperexcitability and tau inhibiting neuronal activity. Amyloid plaques form in the neocortex and spread to deeper brain regions, whereas neurofibrillary hyperphosphorylation occurs in limbic regions and spreads to the neocortex. However, there are studies that imply that, while both pathogenic alterations in the neurological system of AD patients may be recognized, their influence on the disease process and the relationship between the two should be highly complicated, and may even result in two entirely different neuronal actions [11]. Many studies have indicated that abnormal Aβ accumulation can interfere with the GABA inhibitory interneuron function, which can cause aberrant neuron activities and cognitive dysfunction in animals. In addition, it can reduce synaptic transmission and overactivity of neural networks, causing cognitive impairment in vivo. Furthermore, the current research shows that Aβ peptides inhibit cholinergic neurotransmission [12].

There is evidence that shows the involvement of inflammation in AD, including activated microglia within and surrounding senile plaques; the permeability of immune cells and molecules through the blood–brain barrier (BBB) increases with aging, which leads to the neurodegeneration observed in AD patients [13]. In physiological conditions, the immune response is supposed to be terminated when the stimulating pathogen removed, but this mechanism seems to be altered in AD processing and leads to chronic inflammation.

## 2. Neurotransmitter Abnormalities and Cognitive Dysfunction

Neurotransmitters are endogenous chemical messengers that transmit signals across the synapse (between the neurons) and neuromuscular junctions [1]. Neurons in the CNS can be classified by their different neurotransmitters into cholinergic, monoaminergic, glutamatergic or gamma-aminobutyric neurons. Monoaminergic neurons are further classified into dopaminergic neurons and 5-hydroxytryptaminergic neurons. There is now convincing evidence that many types of neurons contain and release two or more different neurotransmitters called co-transmitters [14]. Neurotransmitters are stored in vesicles within the cytoplasm of presynaptic neurons. Fast inter-neuronal signaling in presynaptic neurons, evoked by their presynaptic action potential, can activate calcium channel (VACC) openings and mediate Ca^2+^ entry into the nerve terminal. After that, the transient intracellular uptick triggers vesicle exocytosis and neurotransmitter release [15]. Neurotransmitters bind to the postsynaptic membrane receptors and transmit signals to next neuron [16]. Changes in the neurotransmitters’ synthesis, storage, transportation and degradation can result in neuronal dysfunction, which contributes to AD-related dementia (Figure 1).

### 2.1. Cholinergic Neurons

Acetylcholine (ACh) is a neurotransmitter that is ubiquitous in both the CNS and peripheral nervous system (Figure 1). The cholinergic system is involved in critical physiological processes, such as attention, learning, memory, stress response, wakefulness and sleep and sensory information [17]. ACh is synthetized from choline and acetyl coenzymes by choline acetylase (ChAT) A in presynaptic neurons, which is then stored in vesicles and broken down by acetylcholinesterase in the synaptic gap. The neurotransmitter is transported by the vesicular acetylcholine transporter (VAChT) from the cytosol into synaptic vesicles [18]. ChAT’s activity is controlled by neuronal depolarization and the influx of Ca^2+^, activated by the different protein kinase phosphatases. Ach can be delivered to synaptic gaps and synaptic vesicles via VAChT. PKC can phosphorylate VAChT and regulate its vesicle localization [19] to help deliver acetylcholine to the synaptic gap. VAChT can attach to acetylcholine receptors in the postsynaptic membrane, where it is quickly removed from the receptors [20]. Acetylcholine receptors are excitatory receptors that are classified into the following two types: nicotinic acetylcholine receptors and muscarinic receptors. Nicotinic ACh receptors are ion-gated receptor channels that are selective for cations. Following nicotinic receptor activation, a rapid cellular response is generated. Muscarinic receptors are G protein-coupled receptors that are capable of modulating a wide variety of ion channels. Acetylcholine binding to distinct subtypes of muscarinic receptors stimulates different downstream signaling pathways, producing postsynaptic stimulation or inhibition [21]. Cholinergic neurons can be found mostly in the limbic system of the CNS, such as the hippocampus, amygdala, and hypothalamus, and play a key role in neuronal growth and synaptic plasticity. Cholinergic antagonists can block human use-dependent plasticity and are facilitated by acetylcholinesterase inhibitors [22].

Brain tissue from AD patients shows the reduction of ChAT, acetylcholine synthesis, choline uptake and release, which are markers of cholinergic neuron degeneration and cholinergic neurotransmission [23]. Cholinesterase inhibitors increase the availability of acetylcholine at synapses in the brain and are one of the few drug therapies that have been proven to be clinically useful in the treatment of AD dementia, thus validating the cholinergic system as an important therapeutic target in the disease [17]. One of the first theories that was proposed to explain the etiology of AD was the cholinergic hypothesis. A multitude of cognitive processes, including exploration, rapid eye movement sleep, learning, and memory, depend on the processing of the information needed by Ach.

The results of positron emission tomography demonstrated that nicotinic choline receptors were dramatically diminished in the brains of AD patients, along with the severe loss of cholinergic neurons and decline in ChAT activity and acetylcholine neurotransmitter production [24]. Cognitive impairment is caused by a decrease in the body’s supply of acetylcholine and disruption of the central cholinergic nerve system. Nearly all the patients’ cortexes contained abnormal cholinergic neurons. Recent research implies that the loss and dysfunction of cholinergic neurons are the initial signs of AD pathogenesis and that these neurons are particularly vulnerable to amyloid accumulation [25]. Insufficient amounts of Ach in the synaptic cleft can result from the death of cholinergic neurons and decrease in ChAT activity. Changes in the acetylcholine levels in the synaptic cleft can also directly affect Ca^2+^ levels in postsynaptic neurons because nicotinic receptors in the hippocampus are extremely permeable to calcium. Ca^2+^ influx encourages postsynaptic neuron excitement and accelerates synaptic plasticity, but the mechanism by which Ca^2+^ regulates synaptic plasticity is not yet fully understood [26].

Cholinesterase inhibitors and cholinergic receptor agonists are two types of medications that have been created based on the cholinergic hypothesis. Cholinergic neurotransmission is essential to impaired cognitive function in AD and adult-onset dementia disorders. Cholinesterase inhibitors, such as donepezil and galantamine, can stop the breakdown of Ach in the synaptic cleft and raise the amount of Ach [27]. The main drugs currently used for the treatment of AD are acetylcholinesterase and cholinesterase inhibitors (ChE-Is). The first ChE-I licensed for symptomatic treatment of AD was tacrine. The ChE-Is currently available on the market are donepezil [28], rivastigmine and galantamine, as tacrine is no longer in use, due to its hepatotoxicity [29,30,31,32,33,34,35], as we summarized in Table 1.

Lifelong choline supplementation significantly reduced amyloid-β plaque load and improved spatial memory in APP/PS1 mice, which were linked to the decrease in the amyloidogenic processing of APP, the reductions in disease-associated microglial activation, and the downregulation of the α7nAch and σ1 receptors [36]. Memantine was potentiated by acetylcholine agonists, either alone or in conjunction with it, which reversed scopolamine-induced short-term memory deficits both in mono-treatments and in co-administration [37]. Although these drugs are the only available pharmacological treatments for dementia, many controversies exist about the use of cholinesterase inhibitors in AD for their potential risks regarding clinical treatment, including limited effectiveness and adverse events. In the past two decades, most clinical trials have been stopped due to their serious adverse effects and lack of therapeutic efficacy in Phase III [38], and up to 35% of trials are halted due to adverse events. In addition, AChEI medications cannot alter the history of dementia, and they can only delay the cognitive and functional decline [39]. Taking these situations into account, several European countries no longer support drugs for dementia, hoping the researchers and clinicians pay more attention to non-pharmacological approaches to dementia care [40].

### 2.2. Glutamatergic Neurons

Researchers show that there is a link between the glucose and glutamate alterations with age (Figure 1). The downregulation of the glucose utilization reduces the glutamate levels in AD patients. Alterations in cerebral glucose and glutamate levels precede the deposition of Aβ plaques. Glutamate is the most important excitatory neurotransmitter in the CNS, a non-essential amino acid and the major excitatory neurotransmitter synthesized from glucose [42]. The presynaptic terminal contains synaptic vesicles where glutamate is stored and glutamate is released to the synaptic cleft upon electrical stimulation. The postsynaptic compartment contains glutamate receptors, which are transmembrane proteins responsible for transducing the glutamate signal from extracellular space [43]. Glutamate is transported and released by vesicular glutamate transporters. The glutamate transporters-1 (glutamate transporter-1) and glutamate–aspartate transporter are released into the synaptic cleft by astrocytes [44], and a small part of them is re-taken into the presynaptic membrane, where glutamine is synthesized under the action of glutamine synthetase. Glutamine synthesized in astrocytes can be transported to the synaptic cleft through SNAT3/5, and then transported to neurons by SNAT1/7/8, where glutaminase in neurons can convert it into glutamic acid [45].

There are two kinds of glutamate receptors, metabotropic glutamate receptors (mGluRs) and ionotropic glutamate receptors (iGluRs). iGluRs are ligand-gated ion channels that produce excitatory glutamate-evoked currents, while metabotropic mGluRs are G protein-coupled receptors (GPCRs) that control cellular processes via G protein signaling cascades. iGluR and mGluR signaling has been well studied in isolation, which has revealed the astonishing complexity within both systems [46]. AMPA-type receptors and NMDA-type receptors are transmembrane proteins assembled from four large subunits, with each subunit consisting of an extracellular N-terminal and ligand-binding domain, a transmembrane pore-forming domain and an intracellular C tail. The subunits aggregate in the center to form a channel for ion-selective permeability [47]. The majority of excitatory synaptic transmission in the CNS is mediated by AMPA receptors (AMPARs). AMPARs are the first receptors that react to glutamate, mediating fast excitatory transmission by allowing the depolarization of the postsynaptic neuron through the entry of sodium and the exit of potassium [48]. The most common forms of long-term synaptic plasticity at glutamatergic synapses are triggered by the NMDA receptor, which can produce an influx of Ca^2+^ and Mg^2+^, related to the excitement of synaptic plasticity. Hybrid tetramers of five core subunits called kainate-type receptors can also contribute to both the induction and expression of long-term and short-term forms of synaptic plasticity [49].

The damage of glutamatergic neurons is particularly evident in AD brains. Glutamatergic neurons are crucial for memory, synaptic plasticity and neuronal development. Pathological accumulation of Glu makes it a potent neurotoxin. This is in part due to the time-related exposure, overstimulating the postsynaptic response to cause an increase in the entry of calcium into neurons [50]. The constant activation of AMPA and NMDA receptors causes the continuous influx of Ca^2+^ into neurons, which induces calcium overload in in vitro experiments [51]. In contrast to wild-type mice, AD model animals display lower glutamate aminotransferase activity. Glutamate transporter expression was also downregulated in the brains of AD patients. Glutamate and the metabolite N-acetyl-aspartamine were shown to be reduced in the cortex and hippocampus of AD patients, affecting glutamate metabolism and interfering with the repair and production of myelin [52]. At supraphysiological levels in neurons, glutamate may mediate its degenerative effects through the hyperstimulation of the NR 2B subunit of NMDA receptors, causing the dysregulation of the receptor and resulting in the upregulation of apoptotic regulatory proteins [53]. Increased NFT and amyloid deposition in the CA1 area have been linked to the increase in glutamate levels in mouse cerebrospinal fluid. According to the electrophysiological findings of a previous study, LTP was compromised in the experimental group [54]. The hippocampus is the locus of synaptic plasticity and a critical location for the pathological alterations associated with AD [55]. Synaptic Ca^2+^ influx in hippocampal pyramidal neurons is decreased by Aβ oligomers, which accelerate neuronal cell death [56] and affect neuronal synaptic plasticity by reducing dendritic spine density in the hippocampus and impairing cognitive function in AD patients [57].

Clinical antidepressant and anxiolytic medications that operate on glutamate receptors are frequently utilized. These medicines can act on ionotropic receptors such as NMDA receptors or metabotropic receptors [41]. Numerous studies have demonstrated that these two classes of medications can somewhat alleviate cognitive impairments. Memantine is a moderate-affinity, non-competitive NMDA receptor antagonist that is voltage-dependent. Studies have demonstrated that compared to the placebo group, the memantine administration group of AD patients showed dramatically improved cognitive impairment and behavioral abnormalities [58,59]. Memantine injection boosted synaptic transmission in the hippocampus of APP/PS1 mice, encouraged the regeneration of neuronal dendritic spines, and enhanced the mice’s capacity for learning and memory [60]. Ketamine, as a common antidepressant drug, is a non-competitive NMDA receptor antagonist. In six-month-old rats, long-term low-dose ketamine treatment increased tau deposition and neuronal activity in the prefrontal cortex and PV interneuron of the hippocampus [61]. The intraperitoneal administration of mGluR agonists to mice improved their hippocampal synaptic plasticity by regulating endogenous cannabinoids to activate GABAergic neuron activity [62], and promote LTP in the mouse brains. The administration of mGluR antagonists can improve the performance of mice in water mazes and new object recognition studies and lower amyloid accumulation, which promotes cognition [63].

### 2.3. GABAergic Neurons

The primary inhibitory neurotransmitter in the CNS is GABA (Figure 1), which is crucial for the temporally precise activity of neuronal circuits and synchronized oscillatory activity of neuronal populations [64], and is synthesized through the decarboxylation of glutamate by glutamic acid decarboxylase (GAD). GAD exists in two isoforms, GAD65 and GAD67, which have different molecular weights (65 and 67 KDa), catalytic and kinetic properties, and subcellular localizations. The majority of the glutamate required for GABA synthesis comes from astrocytes [65]. Astrocytes are crucial to the enrichment, synthesis and degradation of GABA, which is taken up by the surrounding astrocytes, transformed into glutamine by the astrocyte-specific enzyme glutamine synthetase and released into the extracellular space from which it is retaken by the neurons and transformed back to glutamate by phosphate-activated glutaminase [66]. Synthetic GABA is concentrated in vesicles by the γ-aminobutyric acid transporter (GAT), stored in the presynaptic cleft and released into the synaptic cleft upon depolarization and Ca^2+^ influx. The GAT is one of the main GABA transporters in the CNS, which principal physiological role is to retrieve GABA from the synapse and transport it to the neurons and astrocytes, thus swiftly terminate the neurotransmission [67]. GABA in the synaptic cleft is broken down by being recycled into presynaptic neurons or being taken up by the surrounding astrocytes, where it is broken down to succinate by the combined action of GABA transaminase and succinate semialdehyde dehydrogenase and enters the tricarboxylic acid cycle or is resynthesized into glutamate with acetyl coenzyme and oxaloacetic acid.

The GABA effects are linked to two different types of GABA receptors, ionotropic and metabotropic GABA receptors, which can be found on the postsynaptic membrane. Upon interaction with GABA, the ligand-gated chloride ion channel of the ionotropic GABA receptor opens, resulting in the Cl- influx and depolarization of neurons. Ionotropic GABA receptors in the CNS are key players in signaling between neurons and can regulate phasic and tonic inhibition in neurons [68]. The G protein-coupled receptor family includes metabotropic GABA receptors, which activate G proteins after binding to GABA and control intricate downstream chemical networks. Metabotropic GABA receptors are distributed in the presynaptic and postsynaptic regions. Presynaptic metabotropic GABA receptors can activate G proteins to bind Gα to Gi/o, which lowers the cAMP levels. In contrast, Gβγ subunits can induce the influx of Ca^2+^ by directly binding to voltage-gated Ca^2+^ channels, which can stop both vesicle fusion and neurotransmitter release. In addition, postsynaptic metabotropic GABA receptors can also activate Gβγ, which inhibits the release of Ca^2+^-dependent neurotransmitters by directly binding to voltage-gated Ca^2+^ channels. The released Gβγ can also directly open G protein-activated inward rectifying K^+^ channels (GIRK), shunting excitatory currents, producing slow inhibitory postsynaptic potentials (IPSP), as well as back-propagating action potentials (APs) to interrupt excitation transmission [69].

The GABAergic system plays an important role in regulating the ratio of neuronal excitation to inhibition as a crucial inhibitory neurotransmitter in the CNS [69]. Alzheimer’s dementia has been linked to GABAergic system disorders, including altered GABAergic receptors and the malfunction of GAD enzymes. A loss of GABAergic neurons in the cerebral cortex, decreased GAD enzyme activity in the hippocampal neurons and increased concentration of GABA in cortical astrocytes were all discovered in AD autopsies. According to in vitro research, the downregulation of GAT3 expression might prevent astrocytes from reabsorbing GABA and disrupt the GABA metabolism, which may cause an imbalance between excitatory and inhibitory transmission in AD patients’ brains and pathological modifications caused by the disease [70]. GABA was markedly downregulated at both the protein and mRNA levels in the hippocampus and cortex of six-month-old APP/PS1 mice, and the density of GABA receptors was reduced near amyloid deposits [71]. hTau downregulated the local level of GABA, presumably by dysregulating GABA synthesis through suppressing GAD67 phosphorylation and inhibiting GABA’s production. Insufficient GABAergic inputs could result in dendrite differentiation and integration impairments of immature neurons, which affect the excitation and inhibition of local neurons in specific functional brain areas, and hippocampal neuronal production [72]. Since the rate-limiting step in GABA biosynthesis is the decarboxylation of glutamate by GAD, it is important to understand how GAD is regulated [73]. Researchers have extensively demonstrated that GAD autoimmunity interferes with GABAergic synaptic transmission. The decreased synthesis of GABA in GAD65 knockout mice can cause GABAergic transmission, which would lead to neuronal hyperexcitability [74].

Gamma oscillations rely on GABAergic inhibition to balance excitation and control spike timing [75]. The primary neuron spikes and neural inhibition can be synchronized by GABAergic neurons. Distinct GABAergic neuron subtypes are involved in mediating rhythmic oscillations in different ways. The primary cells for coordinating the excitation and inhibition of pyramidal neurons include PV^+^ neurons with a unique basket shape and CB1^+^ neurons [76]. Studies have revealed that the number of PV^+^ neurons in the CA1 and CA3 areas of the hippocampus increased in three-month-old APP mice comparing to the WT mice, which might be a result of the diminished inhibitory role of GABAergic neurons with the increase in inhibitory neuron numbers as a compensation mechanism [77]. Additionally, tau fiber tangles in transgenic mice lessen theta and gamma wave rhythm oscillations [78]. Enhancing the gamma rhythm oscillation of AD model animals with light/acoustic stimulation and other techniques can lessen tau protein fibrillary tangles and improve cognitive impairment [79]; studies have shown that after 14 days of transcranial alternating magnetic stimulation in APP mice, these improvements were observed. Flicker stimulation at 40 Hz reduced Aβ in multiple mouse models, including 5 × FAD, APP/PS1, and WT mice, and reduced phosphorylated tau staining in a mouse model of tauopathy, TauP301S, showing that the protective effects of gamma stimulation can be generalized to other pathogenic proteins [80]. The rhythmic oscillations of theta and gamma rhythmic oscillation in mice were restored, and the activity of GABAergic neurons was increased [81]. 

By influencing GABAergic neurons, drugs can effectively treat epilepsy, sleep problems, and other symptoms. Pharmacological modulation of synaptic or extra-synaptic GABAergic signaling mediated by GABAA and GABAB receptors could restore pyramidal neuronal inhibition to normalize aberrant cortical and hippocampal neuronal oscillations in schizophrenia patients. This could ameliorate cognitive impairments such as episodic memory, working memory and executive function in schizophrenia and other neuropsychological disorders [82]. It has been demonstrated that a number of GABAergic medications have specific therapeutic effects on cognitive impairment in AD patients, such as homo lignin, a GABA analog, which can influence GABA receptors. According to various studies, homo lignan can prevent the buildup of amyloid and reduce mild to moderate cognitive dysfunction symptoms of patients. A GABA receptor allosteric modulator can improve cognitive dysfunction in AD patients when combined with other medications. Thus, neurons are activated at the transcription level of the GABAA receptor. In a previous study, it was reported that after using a low-dose GABA receptor allosteric agonist such as lorazepam, the elderly study group had enhanced brain signal variability shown by functional magnetic resonance imaging (fMRI), which was more similar to that reported for younger people [83].

### 2.4. Monoaminergic Neurons

Monoamine neurotransmitters are a subclass of small compounds that cannot cross the blood–brain barrier and are produced through a straightforward metabolic process by rate-limiting enzymes, using amino acids as substrates. In contrast to the quick neurotransmission mediated by GABA and glutamate, monoamine neurotransmitters often act as metabotropic receptors and move more slowly.

The majority of monoaminergic neurons in the brain are 5-HT neurons (Figure 1). A small amount of 5-HT can also be created in the CNS from the substrate tryptophan, although the majority of 5-HT is produced in the gastrointestinal tract. Enterochromaffin cells express tryptophan hydroxylase-1 (TPH1), whereas neurons express TPH2. Tryptophan can be transformed by these two enzymes into the intermediate molecule, L-5 hydroxytryptophan. L-amino acid decarboxylase converts the molecule into 5-hydroxytryptamine (5-HT), then monoamine oxidase on the mitochondrial membrane breaks it down into 5-hydroxyindoleacetic acid [84]. There are seven types of 5-HT receptors, of which six types are G protein-coupled receptors, while the 5-HT3 receptors are ligand-gated ion channels. In addition to indirectly opening G protein-coupled inwardly rectifying K^+^ channels (GIRKs) to hyperpolarize neurons and produce neuronal hyperpolarization, 5-HT1 receptors can activate G proteins and modify downstream signaling cascades. The remaining metabotropic 5-HT receptors shut potassium channels on the basis of activating G proteins, to cause neuronal depolarization and inhibit the opening of voltage-gated Ca^2+^. Ionotropic 5-HT receptors are selectively permeable to K^+^, Na^+^ and Ca^2+^ after interacting with ligands, which can activate neurons [85]. 5-HT receptors are widely distributed in the cortex, thalamus and hippocampus, and ionotropic 5-HT receptors are also distributed in the periphery [86].

Dopamine (DA) neurons are also monoaminergic neurons, which are distributed in the cortex and striatum to regulate motor function, motivation and drive, and cognition. As is the case with 5-HT, DA neurons are produced by the sequential hydroxylation and decarboxylation of tyrosine, and can also be synthesized indirectly from phenylalanine. DA synthesized at the presynaptic terminal of neurons is loaded into synaptic vesicles by monoamine transporter 2 (VMAT2/SCL18A2) [87] when the presynaptic neurons are excited, which is mainly caused by Ca^2+^ influx, leading to neuronal depolarization. Changes in membrane potential cause exocytosis and release into the synaptic cleft, and after the end of the action, they are mediated by dopamine transporters (DAT) or monoamine transporters (MAT), taken back to presynaptic neurons and degraded by monoamine oxidase on the mitochondrial membrane [88]. DA receptors belong to G protein-coupled receptors, which can produce different downstream effects when combined with different GPCRs, and can be roughly divided into D1 receptors and D2 receptors. D1 receptors, coupled to Gαs, activate adenylyl cyclase, generate higher levels of second messenger cAMP, and enhance the activity of protein kinase A (PKA); D2 receptors, coupled to Gαi, inhibit adrenal glycyl cyclase and reduce the intracellular cAMP concentration, thereby inhibiting PKA activity. Dopamine receptors can also catalyze the production of inositol triphosphates (IP3) and diacylglycerols (DAG) by coupling Gαq through regulating phospholipase C, thereby increasing the intracellular calcium levels and activating protein kinase C (PKC), which are involved in the regulation of various signaling pathways. They can also participate in the regulation of other neurotransmitter systems by forming complexes with other neurotransmitter receptors, such as GABA receptors [89].

AD patients exhibit 5-HT neuron loss and reduced 5-HT levels in the cerebral cortex. The levels of 5-HT in the cortical, limbic, sensory, motor, striatal, and thalamic regions of the brain in AD patients with severe cognitive impairment are also lower, coupled with decreased levels of transporter proteins. In addition, 5-HT receptors also couple with GABA receptors, glutamate receptors and acetylcholine receptors to form heterodimers and interact with other neurotransmitter networks [90].

About 50% of AD patients show varying degrees of depressive symptoms, and autopsy results show that the numbers of 5-HT neurons in the brains of AD patients are reduced [91]. The loss of dopaminergic neurons also occurs in the hippocampus and prefrontal regions of AD patients, and the use of dopaminergic antidepressants significantly alleviates the depressive symptoms of AD patients [92]. Using monoamine oxidase inhibitors reduced the level of reactive oxygen species in neurons, avoided oxidative damage, and improved cognitive dysfunction in AD patients [93]. 5-HT receptor-targeted drug can improve the cognitive function in AD rat models. The 5-HT selective reuptake inhibitor fluoxetine can inhibit amyloid deposition and reduce neuronal damage [92]. In vitro experiments show that citalopram, a 5-HT selective uptake inhibitor, reduced Aβ amyloid deposition and APP precursor protein levels in APP gene-overexpressing cells, and improved synaptic protein expression levels [94]. Clinical experiments showed that citalopram improved behavioral patterns in AD patients. 5-HT receptor agonists ameliorated scopolamine-induced spatial learning deficits and increased Ach levels in vivo [95].

### 2.5. Other Factors Affecting Neurotransmitter Synthesis and Release

Animals and humans with AD frequently display morphologically aberrant mitochondria in their brains. Amyloid deposition and tau protein fibrillary tangles, which are the hallmark neuropathological markers of the illness, as well as the AD risk gene (Apo E), cause mitochondrial abnormalities, which lead to pathology alterations and possibly mitochondrial metabolic malfunction (Figure 1) [11]. In severe circumstances, modifications to the nervous system’s metabolic pathways might result in neuronal death, which impairs cognition. The neurons’ synaptic terminal vesicles and cell membranes are fused together by Ca^2+^, which affects neurotransmitter activity in two ways [96]. Neurotransmitter synthesis, processing, modification and secretion all demand a significant amount of energy, which mitochondria must provide. Additionally, concentration fluctuations and mitochondrial activity are associated with one another [97]. Mitochondrial abnormalities have been reported in many different neurodegenerative illnesses besides AD (AD). The activity of several oxidative energy-supplying enzymes decreases, including the pyruvate dehydrogenase complex and ketoglutarate dehydrogenase complex, which affects the mitochondria’s ability to provide energy for the production of neurotransmitters [98]. Additionally, mitochondrial metabolites can take part in the production of neurotransmitters. Furthermore, mitochondria control synaptic vesicle release and neuronal Ca^2+^ concentration by regulating changes in mitochondrial permeability to Ca^2+^. Cyclosporine blocks the Ca^2+^ channels on mitochondria and impairs synaptic plasticity [99]. Drugs that target mitochondria are largely centered on oxidative stress and apoptosis pathways [100], such as MitoQ, which has been proven to reduce tau and Aβ accumulation and synaptic loss in AD mice. Other drugs, for example, MitoTEMPO and MitoApo, are all ubiquinone derivatives, which can preserve neuron function by detoxifying ROS.

## 3. Conclusions

Neurotransmitters are crucial for the survival and upkeep of the physiological functions of neurons. When the neurotransmitter system is compromised, neurons no longer carry out their typical physiological tasks and this also has an impact on synaptic plasticity and cognitive function. Pathological alterations in aberrant neurotransmitter activity or metabolism occur during the development of AD disease, including the loss of cholinergic neurons, dysfunction of glutamatergic neurons, decreased GABA levels and loss of monoaminergic neurons and decreased monoamine levels. Memantine, donepezil, galantamine, and other medications that target the neurotransmitter system are utilized in the clinical treatment of AD. The patients’ cognitive impairment can be greatly improved by this medicine combination. Other neurodegenerative illnesses, such as Parkinson’s, epilepsy and others, are also frequently associated with abnormalities of the neurotransmitter system.

Neurotransmitters have a crucial role in the CNS and they are widely distributed in the CNS. As a result, understanding neurotransmitter production, transport, and metabolism, as well as the control of neurotransmitter networks, helps us to identify new therapeutic targets and investigate the processes underlying the aberrant neurobehavior of AD. This review brings into focus the contribution of neurotransmitter receptors to the pathogenesis of cognitive impairment. We describe the possible mechanisms associated with how neurotransmitters work during the pathological process of AD, in which these changes in neurotransmitters play a compensatory role. In addition, we note the relationship between gamma oscillation in GABAergic neurons and cognitive impairment in AD, which may lead us to the development of an innovative therapeutic method to prevent cognitive decline.

## Figures and Tables

**Figure 1 ijms-24-03841-f001:**
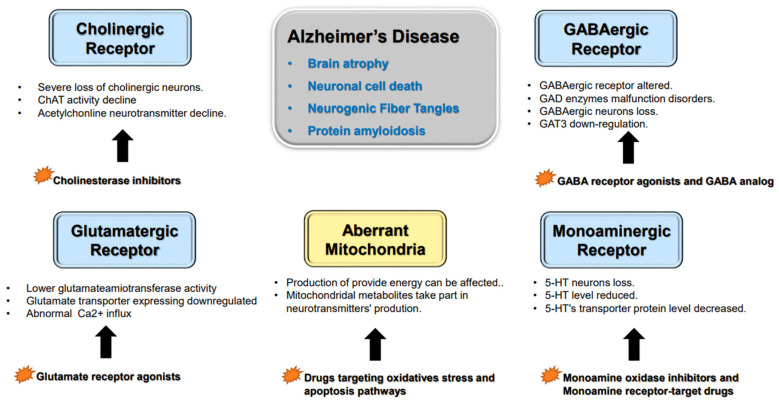
Receptors’ abnormalities and aberrant mitochondria in AD. There are several types of neurons that play important roles in AD. Cholinesterase inhibitors can raise the Ach level. Glutamate receptor agonists could improve the symptoms of AD mice. Researchers have proven that monoamine drugs and inhibitory GABAergic neuron medications can improve cognitive dysfunction in AD patients. Aberrant mitochondria also affect neurotransmitter synthesis and release, and drugs that target oxidative stress and apoptosis pathways reduce AD symptoms.

**Table 1 ijms-24-03841-t001:** Cholinesterase Inhibitors’ Pharmacotherapy in AD patients.

Author	Type	Study Design and Patients		Dosage	Conclusion
		Number of Patients	Group		
Stephen Z. Levine et al., 2021 [26]	Donepezil	2191 AD patients	1339 female and 852 male	Control group: 760 and experimental group: 1431	Logistic modeling showed that donepezil compared to placebo was significantly positively associated with membership in the improvers class, and negatively with high scorers.
Jianping Jia et al., 2019 [27]	Donepezil	241 AD patients	Mild to moderate AD	Donepezil 5 mg/day for at least 4 weeks. All patients received donepezil 10 mg/day for 20 weeks	Single-arm, prospective, multicenter trial. Donepezil 10 mg/day treatment can be tolerated and is also effective in Chinese patients with mild-to-moderate AD, and thus can be used to treat these patients when their response to donepezil 5 mg/day treatment diminishes.
Litao Wang et al., 2021 [25]	Donepezil	90 AD patients		Control group (CG) and experimental group (EG). Patients in CG received donepezil hydrochloride treatment, and on this basis, those in EG received additional RES treatment	Compared with the CG after treatment, the EG obtained significantly higher rates, MMSE scores, and FIM scores (*p* < 0.05) and evidently lower clinical indicators and ADAS-cog scores (*p* < 0.001), and between the CG and EG, no obvious difference in the total incidence rate of adverse reactions was observed after treatment (*p* > 0.05). Conclusion: combining RES with donepezil hydrochloride has significant clinical efficacy in treating AD, which can effectively improve patients’ inflammatory factor indicators, promote their cognitive function, and facilitate patient prognosis.
M. Gaudig et al., 2014 [30]	Galantamine	75 patients	55% women; mean ADAS-cog: 22.3; mean age: 70.2 years	Total daily dose of 24 mg galantamine at final visit	ADAS-cog/11, Bayer-ADL scale (self- and caregivers’ ratings), 10-item NPI and CGI-change, safety and tolerability measures. Galantamine was generally safe and well tolerated during the 3-year observation period. Cognition, behavior, and activities of daily living improved during the 12 months of treatment. At the 3-year follow-up, worsening of all outcomes was measured; however, cognition remained improved compared with an untreated population.
U Richarz et al., 2014 [30]	Galantamine	661 AD patients; 554 were assessed for efficacy		Patients with mild-to-moderate AD received flexible dosing of galantamine (16–24 mg/day) during this study	Galantamine was regarded as generally safe. Importantly, this study revealed that galantamine improved cognitive function above the predicted level in 70% of the patients.
Martin R. Farlow et al., 2013–2015 [31,32,33,34,35,36,37,38,39,40,41,42]	Rivastigmine	1014 patients		716 were randomized to 13.3 mg/24 h (N = 356) or 4.6 mg/24 h (N = 360) patch group	Severe Impairment Battery (SIB) and AD Cooperative Study Activities of Daily Living scale, Severe Impairment Version and ADCS. The 13.3 mg/24 h patch demonstrated superior efficacy to the 4.6 mg/24 h patch on SIB and ADCS-ADL-SIV, a without marked increase in AEs, suggesting the higher dose patch has a favorable benefit-to-risk profile in severe AD.
	Rivastigmine	1014 patients		716 were randomized to 13.3 mg/24 h (N = 356) or 4.6 mg/24 h (N = 360) patch group	A significant therapeutic effect of the high- dose rivastigmine patch on ADCS-CGIC response was observed. The 13.3 mg/24 h patch was identified as a predictor of “improvement” or “improvement or no change”. Patients with minimal worsening/improvement/no change after treatment initiation may be more likely to respond to the treatment following long-term therapy.

## Data Availability

Data available in a publicly accessible repository.

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
