# Peer review of "Neurotransmitters in Prevention and Treatment of Alzheimer’s Disease"

_ijms, 2023, doi:10.3390/ijms24043841_

Round 1
Reviewer 1 Report
ijms-2187622: Neurotransmitters in Prevention and Treatment of Alzheimer’s Disease
This is a comprehensive review, exploring Alzheimer’s Disease (AD) from the point of the aberrant function of neurotransmitters. Although one of the amyloid decreasing therapies, lecanemab has been recently approved by FDA, the so-called amyloid hypothesis of AD is still very controversial. The effects of other antibodies against Abeta are not very efficient. The molecular mechanisms of AD with less relation to Abeta should be further examined. The malfunctions of neurotransmitters are rather a classical point of AD, including the treatment of AD by cholinesterase inhibitors. However, the further research on this matter is still worth being reevaluated. The present review will be very helpful to reconsider the neurotransmitters in AD. The manuscript is well-written and almost ready for publication. The following suggestions may help to increase further the value of this review.
<Major Points>
(1) The clinical effectiveness of cholinesterase inhibitors
There are many controversies about the effectiveness of cholinesterase inhibitors on AD, see, for example, the reference below. This matter may be discussed.
France removes state funding for dementia drugs. BMJ 2019; 367 doi: https://doi.org/10.1136/bmj.l6930 (Published 30 December 2019). Cite this as: BMJ 2019;367:l6930
(2) The relationship between Abeta and each neurotransmitter.
Since lecanemab has been approved by FDA, the relationship between Abeta and each neurotransmitter should be explained individually. In addition, the important point is whether the malfunction of the neurotransmitters is the result of Abeta inducing neuronal degeneration or the results of Abeta-unrelated mechanisms.
<Minor Points>
(a) The paces between lines
There are differences among the spaces between lines, for example those of line 85-89 and those of line91-94. This should be corrected.
(b) Since Fig. 1 is too small, it should be enlarged.
(c) There seems to be some typing errors in line 250-252.
250 acetyl aspartamine were shown to be reduced in the cortex and hippocampus of AD patients, af-
251 fecting glutamate metabolism and interfering with
251 the repair and production of myeli [53]. At supraphysiological levels in neurons, glutamate 252
myeli [53] may be myelin.
(d) author affiliation: line 4-7
4 Zhenqi YANG 1, Yong ZOU 2 and Lifeng WANG 2, *
What is 2_
1 Beijing Institute of Radiation Medicine, 27 Taiping Road, Beijing 100850, China; (Z.Q, Y), yangzq0242@163.com; (Y, Z) tjuzouyong@163.com
* Beijing Institute of Radiation Medicine, 27 Taiping Road, Beijing 100850, China: (L. F, W) fangchang_14@163.com; Tel.: 86-15701120923
What are (Z.Q, Y), (Y, Z), and (L. F, W)?
* should be the corresponding author. Since all the authors belong to the same institute, *address is unnecessary.
Author Response
Dear Reviewer,
Thank you for your letter and for the reviewers’ comments concerning our manuscript entitled “Neurotransmitter Aspects of Prevention and Treatment in Alzheimer”. Those comments are all valuable and very helpful for revising and improving our paper, as well as the important guiding significance to our researches. We have studied comments carefully and have made correction which we hope meet with approval. The main corrections in the paper and the responds to the reviewer’s comments are as following:
<Major Points>
(1) The clinical effectiveness of cholinesterase inhibitors
There are many controversies about the effectiveness of cholinesterase inhibitors on AD, see, for example, the reference below. This matter may be discussed.
France removes state funding for dementia drugs. BMJ 2019; 367 doi: https://doi.org/10.1136/bmj.l6930 (Published 30 December 2019). Cite this as: BMJ 2019;367:l6930
Answer: Thanks for your suggestion. We have checked the papers and added the related contents in line 188-198 according to your advice.
(2) The relationship between Abeta and each neurotransmitter.
Since lecanemab has been approved by FDA, the relationship between Abeta and each neurotransmitter should be explained individually. In addition, the important point is whether the malfunction of the neurotransmitters is the result of Abeta inducing neuronal degeneration or the results of Abeta-unrelated mechanisms.
Answer: We have added the related contents in line 83-87.
<Minor Points>
(a) The paces between lines
There are differences among the spaces between lines, for example those of line 85-89 and those of line91-94. This should be corrected.
Answer: We have modified the formatting errors as you suggested.
(b) Since Fig. 1 is too small, it should be enlarged.
Answer: We have enlarged the figure.
(c) There seems to be some typing errors in line 250-252.
Answer: We have corrected the errors in line 243-246 as shown “N-acetyl-aspartamine were shown to be reduced in the cortex and hippocampus of AD patients, affecting glutamate metabolism and interfering with the repair and production of myelin ”.
(d) author affiliation: line 4-7
4 Zhenqi YANG 1, Yong ZOU 2 and Lifeng WANG 2, *
What is 2_
1 Beijing Institute of Radiation Medicine, 27 Taiping Road, Beijing 100850, China; (Z.Q, Y), yangzq0242@163.com; (Y, Z) tjuzouyong@163.com
* Beijing Institute of Radiation Medicine, 27 Taiping Road, Beijing 100850, China: (L. F, W) fangchang_14@163.com; Tel.: 86-15701120923
What are (Z.Q, Y), (Y, Z), and (L. F, W)?
*should be the corresponding author. Since all the authors belong to the same institute, *address is unnecessary.
Answer: We have corrected the information of authors. Yong Zou is also corresponding author, and we have added his email and telephone number.
We greatly appreciate the efficient, professional and rapid processing of our paper by your team. If there is anything else we should do, please do not hesitate to let us know.
Thank you and best regards.
Yours sincerely,
Zhenqi Yang.

Reviewer 2 Report
1. Lines 55-90 were repeated in lines 91-125, please remove the repetition.
2. Please provide examples of aberrants that may influence neurotransmitter synthesis, storage, transportation, and degradation, resulting in neuronal dysfunction and contributing to AD-related dementia (line 139) in the text rather than mentioning them in Figure 1 legend, and add the references used.
3. You must consider the potential limitations or risks of using cholinesterase inhibitors and other mentioned inhibitors to treat AD.
4. Please add the reference used for lines 150-157, 167-187, 211-213.
5. Please mention “Table 1” in the text.
6. Please edit “oxidatives stress” line 147, and “Metabolotropic” in line 303.
7. Please be mindful of spelling and grammar errors.
Author Response
Dear Reviewer,
Thank you for your letter and for the reviewers’ comments concerning our manuscript entitled “Neurotransmitter Aspects of Prevention and Treatment in Alzheimer”. Those comments are all valuable and very helpful for revising and improving our paper, as well as the important guiding significance to our researches. We have studied comments carefully and have made correction which we hope meet with approval. The main corrections in the paper and the responds to the reviewer’s comments are as following:
Comments and Suggestions for Authors
- Lines 55-90 were repeated in lines 91-125, please remove the repetition.
Answer: We have removed the repetition.
- Please provide examples of aberrants that may influence neurotransmitter synthesis, storage, transportation, and degradation, resulting in neuronal dysfunction and contributing to AD-related dementia (line 139) in the text rather than mentioning them in Figure 1 legend, and add the references used.
Answer: We have supplemented the related contents in line 140-144 and line 331-335.
- You must consider the potential limitations or risks of using cholinesterase inhibitors and other mentioned inhibitors to treat AD.
Answer: We have supplemented the potential limitations or risks of using inhibitors in line 188-198.
- Please add the reference used for lines 150-157, 167-187, 211-213.
Answer: We have re-written this part according to the Reviewer’s suggestion in line 119-141, line 152-164, and line 205-206.
- Please mention “Table 1” in the text.
Answer: It has been added in line 175, “as we summarized in table 1.”
- Please edit “oxidatives stress” line 147, and “Metabolotropic” in line 303.
Answer: We have corrected the spelling errors “oxidatives stress” to “oxidative stress” in line 117 and “Metabolotropic” to “Metabotropic” in line 303.
We greatly appreciate the efficient, professional and rapid processing of our paper by your team. If there is anything else we should do, please do not hesitate to let us know.
Thank you and best regards.
Yours sincerely,
Zhenqi Yang.
